# Gold Nanoprobes for Robust Colorimetric Detection of Nucleic Acid Sequences Related to Disease Diagnostics

**DOI:** 10.3390/nano14221833

**Published:** 2024-11-16

**Authors:** Maria Enea, Andreia Leite, Ricardo Franco, Eulália Pereira

**Affiliations:** 1LAQV/REQUIMTE-Laboratório Associado para a Química Verde/Rede de Química e Tecnologia, Departamento de Química e Bioquímica, Faculdade de Ciências, Universidade do Porto, Rua Campo Alegre, 687, 4169-007 Porto, Portugaleulalia.pereira@fc.up.pt (E.P.); 2Associate Laboratory i4HB—Institute for Health and Bioeconomy, Faculdade de Ciências e Tecnologia, Universidade NOVA de Lisboa, 2819-516 Caparica, Portugal; 3UCIBIO—Applied Molecular Biosciences Unit, Departamento de Química, Faculdade de Ciências e Tecnologia, Universidade NOVA de Lisboa, 2819-516 Caparica, Portugal

**Keywords:** gold nanoparticles, molecular detection, nucleic acid detection, gold nanoprobes, colorimetric assay, thiolated oligonucleotides, nucleic acid sequences, disease diagnostics

## Abstract

Gold nanoparticles (AuNPs) are highly attractive for applications in the field of biosensing, particularly for colorimetric nucleic acid detection. Their unique optical properties, which are highly sensitive to changes in their environment, make them ideal candidates for developing simple, rapid, and cost-effective assays. When functionalized with oligonucleotides (Au-nanoprobes), they can undergo aggregation or dispersion in the presence of complementary sequences, leading to distinct color changes that serve as a visual signal for detection. Aggregation-based assays offer significant advantages over other homogeneous assays, such as fluorescence-based methods, namely, label-free protocols, rapid interactions in homogeneous solutions, and detection by the naked eye or using low-cost instruments. Despite promising results, the application of Au-nanoprobe-based colorimetric assays in complex biological matrices faces several challenges. The most significant are related to the colloidal stability and oligonucleotide functionalization of the Au-nanoprobes but also to the mode of detection. The type of functionalization method, type of spacer, the oligo–AuNPs ratio, changes in pH, temperature, or ionic strength influence the Au-nanoprobe colloidal stability and thus the performance of the assay. This review elucidates characteristics of the Au-nanoprobes that are determined for colorimetric gold nanoparticles (AuNPs)-based nucleic acid detection, and how they influence the sensitivity and specificity of the colorimetric assay. These characteristics of the assay are fundamental to developing low-cost, robust biomedical sensors that perform effectively in biological fluids.

## 1. Introduction

Figure 1 presents a timeline of the use of AuNPs for DNA detection, from their origin in the 1980s and 1990s, followed by four decades of developments, to their near future translation into clinical practice integrated into simple and easy-to-read biomedical devices.

Colloidal gold conjugated to biomacromolecules began to be used in the 1980s, in various analytical methods for clinical diagnostics. Leuvering et al. [1] introduced an immunoassay known as sol particle immunoassay (SPIA), which was later adapted by Mirkin et al. in 1997, in their seminal article about the colorimetric detection of DNA [1,2]. Both protein and DNA versions of SPIA rely on two key principles: (1) The typical red color of AuNPs remains unchanged when recognition biomolecules are adsorbed onto individual AuNPs; (2) aggregation of AuNPs leads to a change in color from red to blue or gray and can be used to detect biomolecule binding events. This change can be easily detected either visually or spectrophotometrically [3,4,5]. Early developments of this interesting property of AuNP, in view of their application in bioassays, were focused on the synthetic optimization of Au-nanoprobes, and several oligonucleotide functionalization methods were proposed [2,6,7]. Soon it became apparent that for each application, namely, to obtain the appropriate sensitivity and specificity of each DNA detection assay, many specific factors such as the functionalization method, the use of spacers, and the base composition and length of the oligonucleotide recognition sequence, as well as oligonucleotide density and mismatches/mutation positioning, should be optimized [8,9,10,11,12]. Alongside these advancements, various applications and commercialization efforts are continuously evolving, driving further innovation and integration. The future goal is to create more reliable and efficient diagnostic tools that enhance patient outcomes and advance personalized medicine [13].

## 2. AuNPs Optical Properties and Their Use in Colorimetric Detection

AuNPs have unique optical properties due to the localized surface plasmon resonance (LSPR), imparting an intense color to AuNPs. The LSPR results from a remarkably high absorption and dispersion of light in the visible/near-infrared region of the electromagnetic spectrum. Several factors significantly influence this phenomenon: intrinsic parameters such as size, shape, and interparticle distance, as well as extrinsic parameters, such as changes in the refractive index of the surroundings. As a result, the absorption and scattering spectra of gold nanoparticles can be tuned across the visible and near-infrared regions of the electromagnetic spectrum (Figure 2).

Spherical AuNPs exhibit a single LSPR band whose wavelength redshifts as the particle size increases. Spherical AuNPs display a great variety of synthesis methods resulting in colloids with uniform shapes and low dispersion in size, providing reproducible optical properties [14,15]. The most widely used spherical AuNPs’ synthesis methods, especially for smaller AuNPs as for the 15 nm ones, are based on trisodium citrate as both a reducing agent of gold salt and a capping agent for the growing AuNPs [14]. For larger diameters AuNPs (e.g., 40 nm), the seed-mediated growth method allows precise control over particle size by adjusting reaction conditions [15]. Larger AuNPs also have higher extinction coefficients, being more intensely colored (Figure 2). Non-spherical AuNPs, such as nanorods and nanostars, display multiple LSPR bands due to their distinct shapes, leading to more complex spectra with overlapping peaks. For non-spherical AuNPs, successful synthesis methods were described in the literature for gold nanostars [16], gold nanotriangles [17], gold nanorods [18], or other shapes [19,20].

Interparticle distance has a crucial influence on spherical AuNPs’ optical properties. As AuNPs draw closer, they can reach a critical threshold where their repulsive forces become insufficient to prevent aggregation. This aggregation induces electromagnetic interactions among AuNPs, leading to plasmon excitation. On the UV-Vis spectrum, AuNP aggregation results in a redshift of the LSPR band due to the enlarged size of the aggregates compared to their non-aggregated counterparts (Figure 2). This color change upon AuNP aggregation is the basis for the assays described here. For 15–20 nm spherical AuNPs, their characteristic red color, corresponding to an LSPR centered at 520 nm, transitions to blue, with LSPR broadening and shifting to approximately 600 nm.

Spherical citrate-coated Au NPs are mostly used as a starting point for the development of colorimetric biosensors [21,22,23,24]. They are easy to synthesize with the desired properties and easy to characterize and manipulate. Citrate ions on the surface of AuNPs confer a highly negative surface charge, thus providing high colloidal stability but also promoting a strong interaction with all cationic biomolecules, leading to AuNPs aggregation [24]. Therefore this interaction lacks specificity and sensitivity to be used in the development of useful biosensors. To enhance specificity, AuNPs must be functionalized with a specific ligand that selectively binds to the analyte of interest, for example, a thiolated oligonucleotide complementary to the target oligonucleotide sequence [25].

These probing moieties, consisting of thiolated oligonucleotide (HS-oligo)-functionalized AuNPs, are called “Au-nanoprobes”, and are the bioreceptors of many promising colorimetric biosensors. These biosensors are ideal for point-of-care applications due to their high sensitivity, accuracy, discrimination ability, low cost, and simplicity. Detection relies on a color change from red to blue caused by AuNP aggregation, which is highly sensitive and visible to the naked eye [25,26]. Aggregation may occur spontaneously upon binding of the target, or it can be induced by increasing the ionic strength of the solution, which reduces electrostatic repulsion, promoting AuNP aggregation (Figure 3) [27,28].

A cross-linking variant of this strategy uses two types of Au-nanoprobes, i.e., AuNP functionalized with HS-oligos that are complementary to both terminal target sites [29]. Hybridization of the target sequences with Au-nanoprobes leads to the formation of AuNP-containing aggregates, thus changing the solution color from the original red to blue (positive result). When no hybridizing target is present, the solution stays red (negative result) (Figure 3).

Conversely, the non-cross-linking strategy is based on only one type of Au-nanoprobes. Hybridization of the Au-nanoprobes with a complementary target increases the colloidal stability due to the high negative charge of the target sequence. Upon increasing the ionic strength, positive samples will remain non-aggregated (red), whereas negative samples will aggregate (blue) (Figure 3).

Based on the observation by Li and Rothberg [30] that, at high ionic strength, ssDNA protects unmodified AuNPs from aggregation (red is a negative result), whereas dsDNA does not (blue is a positive result), some colorimetric systems use non-modified oligonucleotides (Figure 3) [30]. This type of assay often shows high variability; thus, the emphasis of this review article is placed on assays using thiolated oligonucleotide, which enhance functionalization efficiency and create stable Au-nanoprobes.

### 2.1. Cross-Linking Assays

In the cross-linking assay, two different Au-nanoprobes are used for the detection of a target nucleic acid sequence [31,32]. The target sequence, if present in the sample, will work as a linker between the two Au-nanoprobes as it contains complementary regions to both sets of the functionalized gold nanoprobes. Specifically, one part of the target sequence will bind to the HS-oligo on one set of AuNPs, and another part of the target sequence will bind to the HS-oligo on the other set of AuNPs. This hybridization causes the two sets of Au-nanoprobes to come into proximity, leading to aggregation of AuNPs and consequently changes in the optical properties of the solution, turning from red to blue. These aggregates can be reversed by raising the sample temperature above the melting point of the Au-nanoprobes–target complex. Conversely, if the DNA target is not complementary to both Au-nanoprobes, no cross-linking occurs, no aggregation is observed, and thus, no changes are observed in the original red color of AuNPs.

Two hybridization approaches have been used: tail-to-tail and tail-to-head, distinguished by the location of the thiol group on the oligonucleotide. When both Au-nanoprobes have HS-oligos with the thiol group located at the same terminal, they hybridize in a tail-to-head configuration. Conversely, when Au-nanoprobes have HS-oligos with the thiol group located at different terminals, hybridization occurs in a tail-to-tail configuration.

In the absence of amplification techniques, the sensitivity of cross-linking aggregation methods is quite limited, in the nanomolar or subnanomolar range. Most molecular biomarkers in healthy individuals are present at picomolar levels, making them undetectable in cross-linking assays. To enhance sensing performance, several signal amplification methods have been developed and integrated with cross-linking aggregation. These methods include enzyme-aided signal amplification (e.g., exonucleases, nicking endonucleases, and polymerases) and chemiluminescent strategies (e.g., hybridization chain reactions and catalytic assembly). With these amplification approaches, the sensitivity has reached the required levels, with picomolar and even attomolar detection limits for effective diagnostic tests [33].

### 2.2. Non-Cross-Linking Assays

In the non-cross-linking approach, a single Au-nanoprobe for the detection of a target nucleic acid sequence is used. This introduces a clear advantage relative to the cross-linking approach, providing a simpler and less expensive test. If a complementary target to the sequence of the Au-nanoprobes is added, hybridization takes place. This hybridization provides a significant shielding effect to the AuNPs, preventing Au-nanoprobe aggregation even at high ionic strengths. As a result, the solution maintains a red color, characteristic of a well-dispersed colloidal solution of AuNPs. Conversely, if the introduced target sequence is not complementary to the HS-oligo on the surface of Au-nanoprobes, hybridization does not occur. In such instances, the shielding effect provided by the un-hybridized HS-oligos is weak and fails to prevent NP aggregation when the ionic strength is increased. This aggregation leads to a shift of the color from red to blue.

### 2.3. Key Differences Between Cross-Linking and Non-Cross-Linking Assays

Key distinctions between cross-linking and non-cross-linking methods extend beyond their opposite colorimetric responses. Various characteristics have been assessed to compare these two approaches. Among these, the time required to yield a response stands out as a significant parameter. Initially, it was widely believed that non-cross-linking offered quicker results compared to cross-linking, typically within minutes rather than hours. However, further research revealed that the quantity of ssDNA targets in the sample plays a pivotal role in determining the response time. While non-cross-linking indeed demonstrated faster response times with a higher concentration of target sequences, the opposite trend was observed with lower target sequence concentrations, rendering the application of a cross-linking method more advantageous. These conclusions were evidenced by Wang et al. who used three different 15 nm spherical Au-nanoprobes each one containing a 15-mer HS-oligos, and corresponding complementary target sequences to compare the rate of color change between the cross-linking and non-cross-linking aggregation assays [29].

Heidari et al. made a direct comparison of the cross-linking vs. non-cross-linking method using 15 nm spherical AuNPs functionalized with 15-mer HS-oligos for the colorimetric detection of unamplified bovine viral diarrhea virus RNA [20]. The limit of detection was lower for the cross-linking method (6.83 ng/reaction) compared to non-cross-linking (44.36 ng/reaction) and also presented a higher specificity with the cross-linking method. On the other hand, the cross-linking method required optimization and controlling of hybridization temperature and presented a slower response compared to the non-cross-linking method [27].

Reproducibility is another important aspect to discuss. While the cross-linking method solely requires the addition of target nucleic acid sequences, the non-cross-linking method requires an additional step of increasing the ionic strength to induce AuNP aggregation, by the addition of salt. The stability of Au-nanoprobes following the addition of salt has demonstrated variability, potentially impacting the reproducibility of this method. Though these properties are crucial for assay design in specific applications, the introduction of additional variables, such as the detection of single nucleotide polymorphisms (SNPs), introduces further layers of necessary optimization.

Many extensive and excellent reviews have been published along with these almost three decades of using Au-nanoprobes for biodetection [34,35,36,37,38,39]. However, there is a gap in the literature regarding the translation of numerous studies on how the physicochemical properties of these systems affect their detection capabilities. Our main objective with the present critical review is to highlight which are the important factors to take into consideration in producing a reproducible, robust, and inexpensive spherical AuNP-based nucleic acid detection assay that can be easily performed in a homogenous, aqueous format. For that, we will review critically the optimization of Au-nanoprobe preparation. Applications for detection and diagnostic aids for infectious (bacterial and virus) and non-infectious diseases, namely, cancer, will also be covered here.

## 3. Optimization of Au-Nanoprobes

A critical aspect of nucleic acid detection is the ability of gold nanoparticles to selectively aggregate in the presence or the absence of the desired target sequence, depending on the method used. Selectivity is mainly imparted by the characteristics of the Au-nanoprobes, the specific sequence of the HS-oligos used, and their accessibility to hybridize with the target sequence. This accessibility for hybridization depends mainly on five factors, namely, (i) diameter of AuNPs; (ii) mode of binding of HS-oligos to the AuNPs surface, defined by the functionalization method; (iii) type and length of spacers between AuNPs and HS-oligos; (iv) composition and length of the recognition sequence; and (v) positioning of mismatches/mutation [8,9,10,40].

### 3.1. Diameter of AuNPs

The diameter of AuNPs is one of the key parameters in the development of robust assays, as the spectral properties, colloidal stability, and binding properties depend on the size and corresponding curvature of the nanoparticles [8,41] (Figure 2). Interestingly, investigations on the ideal diameter for an optimized assay have revealed a strong dependence on the type of assay. For example, Kim et al. used unmodified AuNPs for a label-free colorimetric assay for MERS-CoV using a disulfide-induced self-assembly approach [42]. From all tested AuNPs with 72, 41, 26, and 19 nm in diameter, only the latter and smaller ones were functional in the assay and not irreversibly aggregated. Using a non-cross-linking approach, our research group compared 20 and 35 nm to detect a single nucleotide polymorphism associated with lactose intolerance. The use of 35 nm Au-nanoprobes proved to be more advantageous, since the assay with the 35 nm nanoprobes provides a reduction of 80% and 48% in the amount of gold and oligonucleotide, respectively [5]. Diaz et al. also evaluated the influence of AuNPs diameter (15, 20–25, and 38 nm) in the detection of DNA targets derived from RNA sequences of SARS-CoV-2, using cholesterol-derivatized hairpin-like HS-oligos. Upon hybridization with a complementary target, the hydrophobic cholesterol moiety is exposed, and the Au-nanoprobes aggregate [43]. The authors concluded that AuNPs in the 20- to 25-nm range of size were an ideal choice between smaller AuNPs that present slow detection and larger AuNPs that are unstable and less sensitive at low target concentrations.

### 3.2. Functionalization Method

Most of the functionalization methods for the preparation of oligonucleotide-containing nanoprobes rely on the use of oligonucleotides with a thiol group in one of its terminals. This thiol group has a strong affinity for the gold surface providing a preferential binding site on the oligo, thus leaving the remaining nucleotide sequence available for hybridization with the target sequence.

To achieve an appropriate coverage of the AuNP surface by HS-oligo, it is critical to decrease the electrostatic repulsion between the highly negatively charged oligonucleotides and gold nanoparticles that commonly have a negative surface charge [24,44,45,46]. Several methods have been proposed where decreasing the negative charge of the oligonucleotides allows a more stable binding and loading capacity on the AuNPs [24,46]. A way to do that is through the addition of positively charged ions or molecules (e.g., cationic surfactants, polycations, or divalent cations like Mg^2+^) or using an acidic environment that induces protonation of the negatively charged phosphate groups, reducing the overall negative charge of the HS-oligo [7,22,23,44,45]. This neutralization is a critical step in improving loading, enhancing binding efficiency, preventing aggregation, and ensuring the stability and functional performance of the nanoparticle-HS-oligo nanoprobes in various applications [22,23,24,46]. Other effective strategies to improve functionalization are known, such as the case of freeze-thaw and solvent-based methods, which are based on increasing the local concentration through physical confinement. Six functionalization methods, reported in the literature, have shown successful results: (i) salt-aging, (ii) pH, (iii) freezing, (iv) instant dehydration, (v) microwave heating-dry, and (vi) evaporative drying. Figure 4 shows schemes for each method, which are further explored in the following six subsections.

#### 3.2.1. Salt-Aging Method

The salt-aging method, initially proposed by Mirkin et al., relies on the gradual increase in ionic strength to neutralize the excess negative charge of HS-oligos, aiding their adsorption to AuNPs [7]. Along with this effect, as the ionic strength of the solution increases, the aggregation rate of AuNPs also increases. It is thus necessary to consider both the adsorption rate of HS-oligos and the aggregation rate of AuNPs. If the adsorption of HS-oligos is faster than NPs’ aggregation, stable Au-nanoprobes can be obtained. Otherwise, aggregation of NP will affect the adsorption process, making the process unreliable. To avoid aggregation, usually the ionic strength increases very slowly, over the course of a few days. Another strategy is the use of surfactants, enabling the consistent functionalization of AuNPs of larger sizes up to 250 nm [8]. The use of sodium dodecyl sulfate (SDS), Tween 20, and Carbowax was studied [8]. Protection against aggregation comes from the formation of a surfactant double-layer at the surface of the nanoparticles. In the case of anionic surfactants, both electrostatic and steric stabilization may play significant roles, whereas non-ionic surfactants, like Tween and Triton, likely provide only steric stabilization. Fluorinated surfactants, on the other hand, may form highly rigid double layers, resulting in exceptional AuNP stability. In all cases, the thiol group on the oligonucleotide end can penetrate the surfactant layer and strongly adsorb onto the AuNP surface. However, this penetration process remains slow, typically requiring over 2 h [8].

Despite its extensive application with small NPs (15–20 nm), the application of this method to NPs with diameter ≥ 35 nm, which are more sensitive to the increase in ionic strength, usually leads to severe aggregation. The method also requires a high excess of oligonucleotide to increase the adsorption rate and avoid aggregation during functionalization.

#### 3.2.2. pH Method

In the pH method, pioneered by Zhang et al., instead of gradually increasing the ionic strength of the solution, neutralization of the oligonucleotide is induced by rapidly lowering the pH to 3 [22,23]. At this pH value, protonation of adenosine and cytosine occurs, (pKa of 3.5 and 4.2, respectively), which partially neutralizes the negative charge of HS-oligos [47]. Despite the increase in ionic strength due to the addition of buffer, the adsorption of this partially protonated HS-oligos to the surface of AuNPs is very fast as the pH decreases, effectively shielding the AuNPs from aggregation induced by the increase in ionic strength.

This method has been used with success for both small (15 nm) and larger (40 nm and above) AuNPs. It offers three main advantages over the salt-aging method: (1) superior functionalization efficiency; (2) it can be completed over the course of several hours rather than days; and (3) it was successfully used for functionalization of larger spherical AuNPs [22,48].

When comparing the hybridization efficiency and the average binding affinity to the target DNA, both methods yielded comparable hybridizability, although higher loading capacity was observed for nanoprobes prepared using the low-pH method [49]. However, the low pH approach may be less effective for DNA sequences rich in guanine and thymine, for which protonation requires a lower pH, and thus a higher probability of AuNPs’ aggregation [47]. A concern is also the long-term stability of oligonucleotides at low pH, due to depurination of nucleotides, especially guanine, resulting in the loss of DNA bases and subsequent cleavage of the oligonucleotide strand [46].

#### 3.2.3. Freezing–Thaw Method

In the freezing–thaw method for functionalizing AuNPs, strong adsorption of HS-oligos on AuNPs is induced by freezing, at low salt concentration and high HS-oligo concentration [21,50,51]. During freezing, the formation of ice crystals leads to a concentration of solutes in the remaining liquid phase. This concentrated liquid phase facilitates the adsorption of HS-oligo molecules onto the surface of AuNPs, as it was demonstrated that oligonucleotides are aligned and stretched upon freezing, facilitating fast adsorption [51,52]. However, the freeze–thaw method is constrained by the impact of oligonucleotide secondary structures on the AuNP-based labeling reaction, limiting its sequence universality for the construction of AuNP-based nanoprobes [53].

#### 3.2.4. Instant Dehydration Method

The dehydration method relies on the instant dehydration in butanol (INDEBT) of an HS-oligo/AuNP mixture [54,55]. The removal of water from HS-oligo/AuNPs in the presence of butanol results in the formation of a dehydrated “solid solution”, which significantly speeds up adsorption to nanoparticles via Au–S bonding. Critical is both the dehydration-shrunken DNA size and the reduced double layer charges along with uniform and utmost-concentrated HS-oligo/AuNP “solid solution”. This method is simple, efficient, scalable, and fast producing Au-nanoprobes with record-high DNA density in just seconds. The INDEBT-based functionalization method involves two rapid solution-mixing steps: (a) an aqueous solution of AuNPs and thiolated DNA is introduced into a butanol phase and rapid functionalization occurs during this dehydration process and (b) a freshwater phase is added to rehydrate and collect the newly formed solids [54]. The process is extremely fast (a matter of seconds) compared with previously described techniques including the freezing method and leads to a higher DNA loading [54,55]. This “flash synthesis” method was successfully used in the functionalization of AuNPs with diameters between 5 and 45 nm, along with different oligonucleotide sequences (21, 59, and 89 bp) and even gold nanorods [54]. Nevertheless, while for the low pH and freezing method, oligonucleotide strands can be found stretched and aligned, and this is important in target recognition and diagnosis, for the dehydration process, the effects of butanol on the conformation of DNA remain to be explored [55]. Even more, the method does not allow for variation in the oligo loading, as only high-density loading is possible. It requires a high excess HS-oligo to ensure operation stability, as AuNPs’ aggregation can easily occur upon the addition of the butanol phase. Large-scale production is limited as a copious volume of butanol would be needed with environmental/biological negative impact.

#### 3.2.5. Microwave Heating-Dry Method

The microwave-assisted heating-dry method relies on the use of microwave heating to remove, within minutes, the water from an AuNP-DNA solution and obtain highly stable Au-nanoprobes [56]. The process involves two key steps: (a) heating the aqueous solution to stretch nucleic acid strands and (b) drying the mixture to compress the reaction volume, promoting the adsorption of HS-oligos onto AuNPs [56]. It can be successfully used for HS-oligos but for non-thiolated oligos, the conjugation relies on the presence of poly(T/U) polyT tag on the far end (away from the gold surface) of the DNA strand to enhance functionalization efficiency [56]. The method is completed in a few minutes, which makes it faster than conventional techniques such as salt-aging and pH, and achieves high oligonucleotide density on the surface of AuNPs and stability, even with long-chain or structured oligonucleotides. The method is also more environmentally friendly than the instant dehydration method, as it eliminates the need for toxic chemicals. It can be successfully applied to functionalize spherical AuNPs with diameters higher than 13 nm and even gold nanorods (AuNRs). The resulting Au-nanoprobes show high long-term stability up to 47-day storage and at a wide pH range stability (pH 3~11) for both thiolated and non-thiolated oligonucleotide–AuNP conjugates. The hybridization efficiency of the resulting Au-nanoprobes was tested on a test strip hybridization experiment, and it was found that the oligonucleotides in the probe can hybridize with a complementary target.

#### 3.2.6. Evaporative Drying Method

The evaporative drying method shares similarities with the instant dehydration and microwave heating-dry techniques but relies on a simpler, yet highly effective, evaporation process [57]. This approach has the advantages of dehydration, such as reduced DNA size and diminished double-layer charges, while forming a uniform and concentrated oligo/AuNP solid solution. Importantly, it overcomes the limitations of other methods, including sequence dependence, low oligonucleotide density, and the use of toxic solvents [54,56,57]. The evaporative drying process typically occurs at 60 °C in a centrifugal vacuum concentrator, for 30 min or less. However, it can be performed at different temperatures using a conventional heater, with times ranging from several minutes to tens of minutes, depending on sample volume and heating rate. The method is also scalable with the use of a rotary evaporator.

The functionalization method is extremely simple, scalable for bulk production, and requires minimal precautions. It also presents several other advantages such as enabling thiolated and non-thiolated oligonucleotide functionalization on gold nanoparticles of various sizes (5 to 35 nm), shapes including gold nanorods, and surface-capping ligands as citrate or bis(p-sulfonatophenyl)phenylphosphine dihydrate dipotassium salt (SPP), without requiring specific base sequences or strand lengths (oligonucleotide 21 to 89 mer length) [57]. Also, different heating and evaporation formats can be used, both for small and large solution volumes without significantly impacting functionalization efficiency. It allows fully adjustable oligonucleotide density on the AuNPs and valence-controlled oligonucleotide–AuNP conjugates. Additionally, quantitative preparation is possible without using excess oligonucleotides, and the resulting dried products can be easily stored for long periods under ambient conditions [57].

Table 1 compares these six functionalization methods of AuNPs in terms of their speed, oligo to AuNP density, required equipment, and the obtained Au nanoprobe stability. The latter is the most determining factor in prolonging the shelf-life of assay kits.

While the traditional pH and salt-aging methods are straightforward and accessible, they offer less control over oligonucleotide density and are prone to nanoparticle aggregation. In contrast, more recently developed techniques, such as evaporative drying, microwave heating, and flash synthesis, offer fast and efficient functionalization with higher oligonucleotides to AuNPs densities. These modern methods produce stable Au-nanoprobes with high oligonucleotide densities in record time, making them particularly suitable for applications that require high functionality, such as gene delivery systems or advanced diagnostics.

### 3.3. Use of Spacers

The use of a spacer (Figure 5) between the thiol group and the recognition oligonucleotide sequence is highly recommended, as it relocates the recognition sequence away from the nanoparticle surface, mitigating steric hindrance during target hybridization. This modification also improves DNA loading and avoids direct interaction of the recognition sequence, another factor for steric hindrance during target hybridization. Several spacers that can bind the 3′ end of the 5′ end of a DNA strand have been used, including polyethylene glycol (PEG), or 10–30 bases long sequence of adenine (PolyA) or thymine (PolyT) nucleotides, etc. (Table 2).

The interaction between the spacer and AuNPs is critical for the overall performance of the probe. For example, among nucleobases, adenosine has the highest affinity for the gold surface [44]. Thus, PolyA adsorbs to the surface with a side-on conformation. Therefore, surface coverage is high, avoiding non-specific interactions with other molecules. Nevertheless, in comparison with end-on adsorption (a typical characteristic of HS-oligos), side-on conformation provides poorer oligo loading, potentially decreasing sensitivity. The use of PEG as a spacer allows increased loading relative to PolyA or PolyT. This is due to the neutral charge of PEG, allowing a better packing, relative to the highly negatively charged oligos. In addition, the affinity of the ether groups in PEG towards gold is quite low, thus avoiding adsorption in side-on conformation [8].

Spacer composition has also a strong impact on the long-term functional stability of oligo-AuNPs nanoprobes. For example, the use of a spacer with a high affinity to AuNPs, such as polyA, leads to better long-term functional stability as compared with polyT [58]. Therefore, a polyA tag in the oligonucleotide is widely used either with a thiol functional group in the terminal or without any chemical modification, since polyA can strongly adsorb to the gold surface, providing good stability of the resulting nanoprobes compared with spacers containing other nucleobases [47,52,58,59].

Another important factor is the length of the spacer. A small spacer is usually preferred, most commonly a 10-base-long sequence of thymine or adenine, relative to longer sequences. The preference for the shorter length spacer is associated with better hybridization efficiency, enhancing the sensitivity of detection assays, and resulting in stronger and more immediate signal changes upon hybridization. It can also improve surface density, and it is simpler and more cost-effective. On the other hand, a longer spacer allows for a more stable nanoprobe [9]. A longer spacer will have better flexibility, which will lead to coiling effects of the whole sequence around the AuNP, decreasing the loading capacity [60]. In addition, better performance is observed for nanoprobes where polyA or polyT tags are inserted in the 3′ end. The favorable insert position at the 3′ end is related to the synthesis process, which starts at 3′, assuring correct modification of the sequence [60].

**Table 2 nanomaterials-14-01833-t002:** Examples of spacers that can be used to produce Au-nanoprobes with enhanced hybridization capabilities.

Chemical Configuration	Spacers	Reference
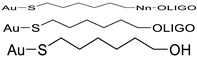	6-MercaptoHexanol-DNA	[61]
	HS-(CH_2_)_6_-5′DNA	[62,63]
	DNA-(CH_2_)_3_-HS-3′	[6,64]
	HS-(CH_2_)_6_-O-(PO_3_)-5′DNA	[65]
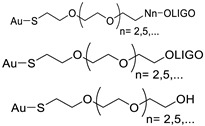	5′-(SS-HEG)_2_-DNA	[66]
	5′-(HS-PEG)-DNA	[8]
	-(CH_2_-CH_2_-O)_3_-5′DNA	[67]
(AAA)n-OLIGO	polyA-DNA	[8,47,52]
(TTT)n-OLIGO	polyT-DNA	[8,49,56]

### 3.4. Composition and Length of the Recognition Sequence

The length of the recognition sequence is a critical parameter for gold nanoprobe-based DNA detection and can influence the sensitivity, specificity, hybridization efficiency, and robustness of the assay. Shorter sequences have the advantage of fast hybridization, but they generally have low stability. In addition, the use of short recognition sequences increases the number of false positives, since the recognition sequence can be partially complementary to other DNAs in the sample. On the other hand, the use of longer sequences provides better Au-nanoprobe stability and increased specificity, but with longer detection times [45,49,56]. The length of the recognition sequence should be selected taking into consideration the requirements of the detection assay, and it must balance the need for speed, sensitivity, and specificity.

The nature of the oligonucleotide sequence used in the nanoprobe influences the stability, specificity, and kinetics of hybridization in DNA detection. The sequences with a high GC content are usually more stable but require longer incubation times for complete hybridization to occur. A low GC content, on the other hand, imparts fast hybridization but less stable Au-nanoprobes. It is also important to avoid the formation of secondary structures to obtain reliable and sensitive DNA detection. The presence of a secondary structure introduces an additional thermodynamic barrier to hybridization, which is especially significant at low ionic strengths [68]. In fact, secondary structures can prevent proper proximity-induced changes, which are crucial for clear signal development. Moitra et al. suggested that other factors that should be taken into consideration when designing Au-nanoprobes are: a GC content of the oligonucleotide between 40% and 60%; the target sequences containing GGGG should be eliminated; the average unpaired probability of the oligonucleotides for target site nucleotides must be ≥0.5; and the binding energy of the oligonucleotides should be compared with the target sequence and the binding energy cut-off for the selection of oligonucleotide has to be kept at ≤−8 kcal/mol [69].

### 3.5. Oligonucleotide Density and Mismatches/Mutation Positioning

The density of oligonucleotides at the AuNP surface affects the efficiency, stability, specificity, and sensitivity of DNA detection assays. A high oligonucleotide density can cause steric hindrance, reducing hybridization efficiency, while low density improves accessibility and hybridization efficiency with the target DNA, but may compromise the stability of the nanoprobe and its duplex [10,70].

Song et al. [71] showed that there are limiting values for oligonucleotide surface functionalization density that determine the color scheme for the non-cross-linking assay [71]. In fact, the most common scheme, at high salt concentration, is blue for negative samples, due to Au-nanoprobes aggregation, and red for positive samples, when a complementary target is present and avoids aggregation of AuNPs (see Figure 3, non-cross-linking panel). Nevertheless, this scheme is only valid for a surface oligonucleotide density of less than 26 pmol cm^2^. In this case, the colloidal stability of the Au-nanoprobes upon salt-induced aggregation is impaired by dsDNA formed at the AuNP surface. Conversely, an oligonucleotide density at the Au-nanoprobe greater than 34 pmol cm^2^ imparts long-term stability to the Au-nanoprobe with red color even at high salt concentration, i.e., the negative assay has a red color, while hybridization with complementary DNA will induce aggregation, and color changes to blue. This latter color change scheme was observed by Maeda and coworkers [72,73], but as it involved a higher oligonucleotide load on the AuNP and increased cost of the associated assays, it was not developed further [72,73].

Seminal work by Demers et al. [6] established a fluorescence-based method to determine the surface coverage and hybridization efficiency of Au-nanoprobes [6]. Using a “salt-aging” functionalization method on 16 nm spherical AuNPs, a surface coverage of hexanethiol 12-mer oligonucleotides of 34 pmol/cm^2^ was obtained. Also, maximal hybridization efficiency with fully complementary targets was observed at a density of 20 pmol/cm^2^ [6]. In line with these findings and using the same methods and the same type of oligonucleotides to functionalize 13.5 nm Au-nanoprobes, Doria et al. showed that the highest hybridization efficiency and discrimination of single-nucleotide polymorphisms at room temperature occurs at a density of 83 ± 4 thiol-oligonucleotides (24 pmol/cm^2^) [10,70]. They also demonstrated that the mismatch should be positioned at the 3′ end of the Au-nanoprobe [10]. In fact, mismatch positions affect the hybridization efficiency, stability, and specificity of DNA detection [11]. Central mismatches significantly reduce duplex stability and enhance discrimination between matched and mismatched sequences, improving specificity. Terminal mismatches have a smaller effect on stability but can still be used for detection under optimized conditions [10]. The effect of mismatch location in the detection assay appears to be closely linked to the morphology of the gold nanoparticles. Specifically, when analyzing a detection method based on the shifts in Localized Surface Plasmon Resonance observed during the incubation of DNA targets with oligo-functionalized gold nanotriangles, our research group concluded that optimal target hybridization is achieved when the mismatch is located at the center of the target [74].

## 4. Applications

Gold nanoprobes can be used for detection in distinct application fields including cancer and infectious diseases, food safety, and environmental detection of biological contaminants [69,75,76,77,78].

Table 3 presents several examples, found in the literature since 2018, of the use of Au-nanoprobes for the detection of specific nucleic acid sequences related to diagnostics, highlighting the main assay characteristics and obtained specificity. In this Applications section, we also mention especially successful applications of Au-nanoprobe-based methods to the diagnostics of non-infectious diseases (e.g., cancer) and bacteria- or virus-caused infectious diseases.

### 4.1. Colorimetric Biosensors for Cancer Detection

The use of biosensors for the development of simple and rapid response sensor devices for cancer detection is still in its development stages [84]. Colorimetric sensors, particularly those using AuNPs, represent a cost-effective alternative to conventional cancer diagnostic tools, as they can have high sensitivity, reduce cost, and are very simple to use, making them commercially viable and accessible to the public [85,86,87]. AuNPs are more expensive than other metal nanoparticles (e.g., AgNPs,), but they present better optical properties, enabling clear visual detection. Gold is also a biocompatible material, a property desired for cancer treatment and diagnostics. This high biocompatibility of AuNPs makes them more desired also due to econanotoxicology concerns [88].

Lee et al. demonstrated that 18 nm spherical AuNPs functionalized with a 20 mer HS-oligos can be used for the detection of EGFR mutations associated with lung cancer [89]. Assah et al. showed that 13 nm spherical AuNPs functionalized by the pH method with HS-oligos can discriminate between wildtype p53 protein and mutated p53 protein, using a cross-linking approach with a limit of discrimination of 5 nM [81]. Dysfunctional mutant p53 proteins are considered a clinically viable target for both diagnostics and therapeutics, as they are associated with cancer development and maintenance [90].

Some aspects need to be improved before Au-based colorimetric biosensors for cancer diagnostics reach their full potential. For example, the type of cancer cannot be accurately determined based on one mutation, as different types of cancer can be associated with the same mutation; therefore, one mutation is not 100% specific to one type of cancer. Therefore, it is necessary to study the multi-modeling of NPs conjugated with multiple DNA-specific sequences. Another challenge is the fact that in metastatic cancer, diagnosing the origin of the cancer is difficult due to its rapid spread.

### 4.2. Colorimetric Biosensors for Bacterial and Viral Infections

Regarding the detection of bacterial and viral infections, AuNPs-based detection methods have been described for the last decades [74,91]. The COVID-19 pandemic gave an important boost to these efforts [92,93]. The advantage of AuNPs over other nanomaterials for the development of an easy-to-produce and reliable point-of-care SARS-CoV-2 immunoassay, positioned them to lead the global market. Also, AuNP-based RNA/DNA detection applications reinforced this dominance in the application of AuNPs to rapid molecular diagnosis. In the study by Moitra et al., 30 nm spherical AuNPs functionalized with 20 mer thiolated antisense oligonucleotides specific to the N-gene (nucleocapsid phosphoprotein) of SARS-CoV-2 can diagnose positive COVID-19 cases within 10 min from isolated RNA samples [69]. These thiol-modified nanoprobes selectively aggregate in the presence of the target RNA sequence of SARS-CoV-2, causing a change in surface plasmon resonance. Additionally, RNaseH cleaves the RNA strand from the RNA–DNA hybrid, resulting in a visually detectable precipitate due to further agglomeration of the AuNPs. The assay demonstrated good selectivity in the presence of MERS-CoV viral RNA, with a detection limit of 0.18 ng/μL of RNA containing SARS-CoV-2.

## 5. Challenges and Opportunities

Translation to the market of AuNPs aggregation-based assays is still limited, despite many key advantages over other assays in the market, including label-free protocols, fast interactions in solution, and the ability to detect results visually or with inexpensive instruments. Possibly the main challenge for AuNP-aggregation-based assays is to ensure reproducibility of the results due to the inhomogeneity of colloidal aggregation. Basic physicochemical studies of these systems are thus an opportunity to deeply understand the underlying mechanisms of functionalized nucleic acid hybridization and aggregation. In these studies, beyond traditional UV-visible spectroscopy, emerging detection techniques have become more relevant, such as electrophoretic light scattering (including DLS and zeta-potential measurements), and nanoparticle tracking analysis (NTA). Also, Surface Enhanced Raman Spectroscopy (SERS) has found applications for enhanced sensitivity in detection, when non-spherical (especially star-shaped) nanoparticles are used in mounting Au-nanoprobes. This brings us to the second important challenge of AuNP aggregation-based assays, which is color detection and multiplexing. Using AuNP with different morphologies, including spherical, alone, or mixed, gives a vast array of color combinations, which can be used for maximizing the color difference upon nanoparticle aggregation. However, different shapes exhibit varied optical properties and reactivity, which can complicate the translation of existing methodologies, and reinforce the need for robust stabilization strategies [94]. As for multiplexing, interesting options have been suggested, for example, using two types of nanoprobes (AuAg-alloy and Au), for simultaneous differential analysis of two sequences on the mandatory assessment of whether a given genotype is present in homo- or heterozygous form. Nevertheless, these types of systems have proven very difficult to work with, due to the introduction of silver, a enough different material from gold to present different functionalization conditions and different colloidal stability of the obtained nanoprobe.

Another challenge of these AuNP-based aggregation assays is related to the direct use of biological samples such as blood, urine, or saliva, as these can interact with the Au nanoprobes and interfere with the colorimetric response [95,96]. For instance, saliva interferes with the sensor in two main ways: (i) by suppressing color change signals due to proteins nonspecifically adsorbing to the nanoparticles and (ii) by blocking aggregation and generating false results due to specific electrolytes that induce aggregation. The authors used protein extraction processes to mitigate these effects [96]. In another example in which the direct use of a biological sample would be helpful by avoiding a prior extraction step, miRNAs as biomarkers in osteoarthritis could only be effectively detected using a cross-linking method, in the patient plasma extracted form, while plasma contents induced undesired AuNP aggregation [97]. Overcoming this challenge is determining to improve the sensitivity and specificity of Au-nanoprobes for hybridizing to their target in a biological matrix, advancing their diagnostic capabilities. This opens an opportunity for the development of new techniques and functionalization strategies that enhance the binding affinity and selectivity of the oligonucleotide-functionalized AuNPs, for which different types of functionalization methods and/or spacers and/or oligonucleotide to AuNPs ratios might be needed. These would in turn impact in pH, temperature, or ionic strength required to influence the Au-nanoprobe colloidal stability and thus the performance of the assay. Cross-linking and non-cross-linking approaches each offer unique advantages and challenges for AuNP-based colorimetric detection. Cross-linking assays, involving two complementary Au-nanoprobes that aggregate in the presence of a target, provide high specificity and can reach attomolar sensitivity with amplification [31,32]. However, they are slower, require strict temperature control, and show reduced sensitivity without amplification. Non-cross-linking assays, using a single Au-nanoprobe, are quicker and simpler, making them ideal for high-concentration targets and point-of-care use, but are sensitive to ionic strength adjustments, leading to variability and lower sensitivity at low concentrations. Careful optimization of parameters like nanoprobe functionalization, salt concentration, and temperature control is essential to ensure reliable performance across both methods [27,29]. Furthermore, detection and interpretation techniques like UV/Vis spectroscopy and nanoparticle tracking analysis (NTA) help to quantify aggregation accurately, boosting result accuracy. By addressing these key elements, researchers can achieve robust, reproducible, and diagnostically relevant results tailored to specific applications, whether in infectious disease detection or broader diagnostic use.

Future research on gold nanoparticle-based nucleic acid detection assays could explore advanced functionalization strategies to improve the stability and selectivity of these nanoprobes in complex biological samples. Incorporating machine learning algorithms to fine-tune nanoparticle properties may also enable color multiplexing and enhance the simultaneous detection of multiple targets, expanding the capabilities of point-of-care diagnostics. The future use of optimized gold nanoprobes for AuNPs-based colorimetric detection of specific nucleic acid sequences holds great potential to positively impact both socio-economic factors and point-of-care testing, improving healthcare accessibility, early disease detection, and personalized and targeted treatments.

## Figures and Tables

**Figure 1 nanomaterials-14-01833-f001:**
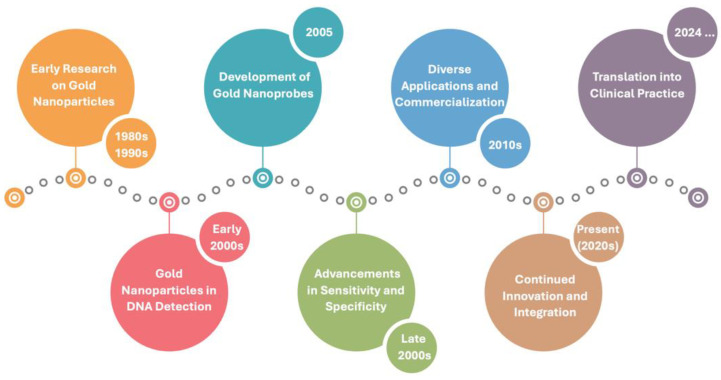
Timeline of AuNPs use for nucleic acid detection.

**Figure 2 nanomaterials-14-01833-f002:**
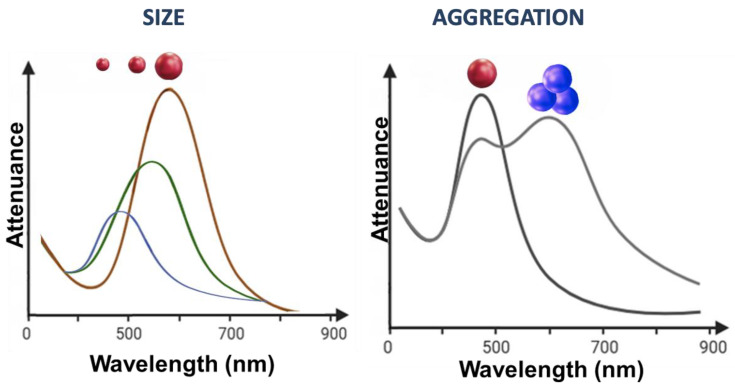
Dependence of LSPR on spherical gold nanoparticles diameter and aggregation state.

**Figure 3 nanomaterials-14-01833-f003:**
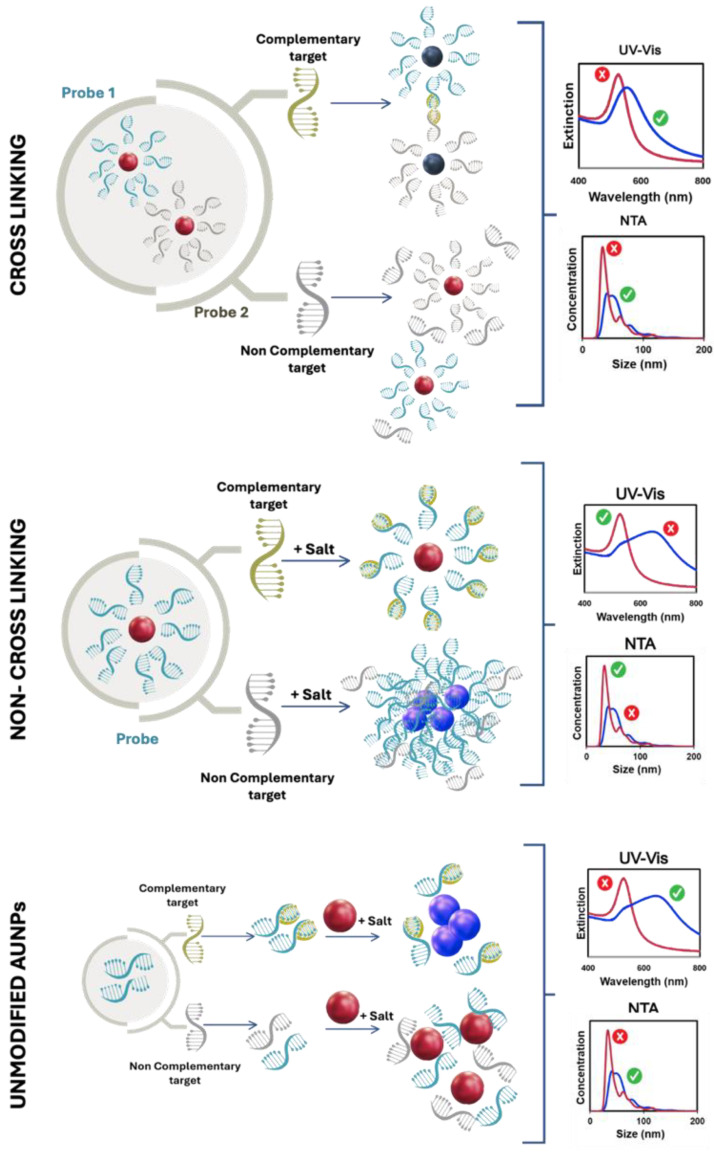
Colorimetric detection methods using spherical AuNPs: (Top panel) Cross-linking assay—a color change occurs as nucleic acid sequence strands specifically hybridize with complementary sequences, reducing the distance between particles, and resulting in a blue solution (positive test). In the absence of complementary sequences, the solution stays red (negative test). (Middle panel) Non-cross-linking assay—an increase in ionic strength induces AuNP aggregation, resulting in a blue solution (negative test). When complementary targets are present, the solution stays red (positive test). (Bottom panel) Colorimetric assay using unmodified AuNPs: In the absence of complementary sequences, only single-stranded DNA (ssDNA) is present, stabilizing AuNPs against salt-induced aggregation, and the solution stays red (negative result). Conversely, when hybridization occurs in the presence of a complementary sequence, double-stranded DNA (dsDNA) forms, and aggregation occurs (blue solution is a positive result). UV/vis spectra and Nanoparticle Tracking analysis (NTA) profiles are shown with blue lines corresponding to aggregated AuNPs samples and red lines to non-aggregated ones. Also indicated are the positive (green check) and negative (red cross) results for each test.

**Figure 4 nanomaterials-14-01833-f004:**
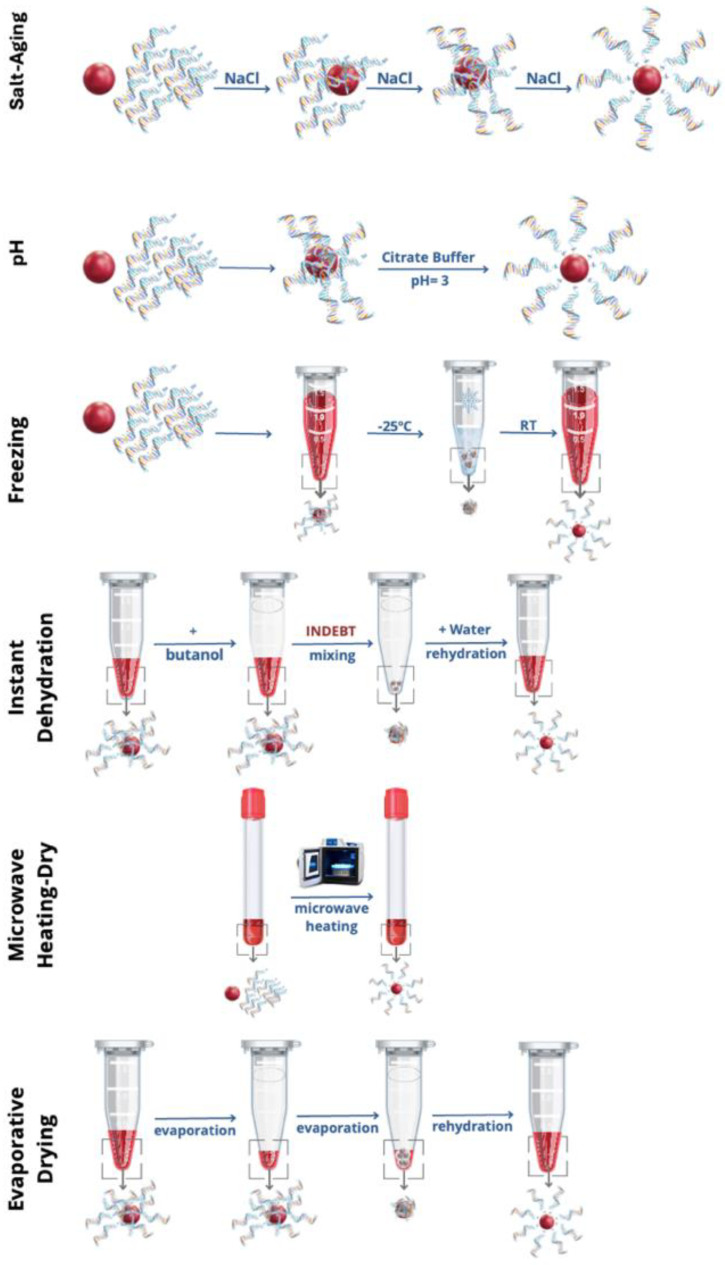
Published successful functionalization methods of AuNPs with HS-oligos, resulting in Au-nanoprobes.

**Figure 5 nanomaterials-14-01833-f005:**
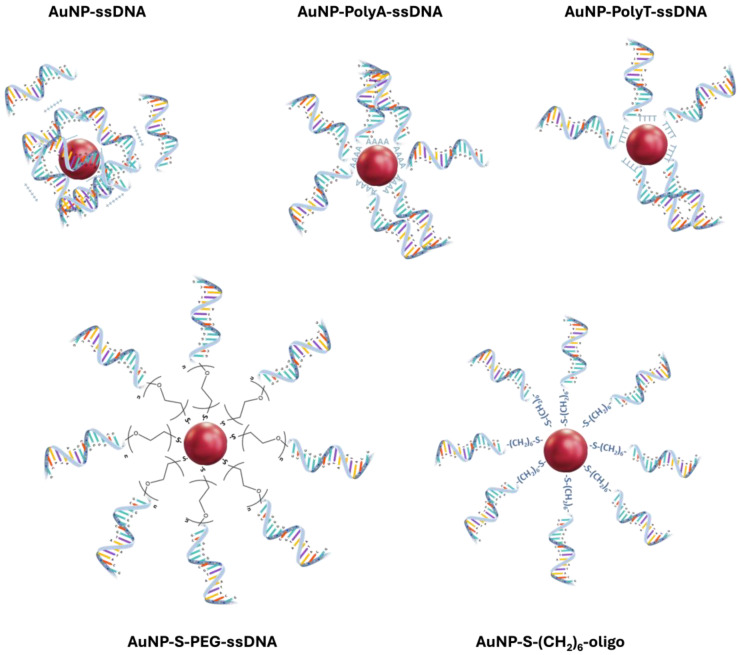
Examples of Au nanoparticle interaction with (i) ssDNA, (ii) PolyA-ssDNA and PolyT-ssDNA, (iii) PEG-ssDNA, and (iv) thiolated-(CH2)6-ssDNA.

**Table 1 nanomaterials-14-01833-t001:** Comparison of functionalization methods of AuNPs.

Functionalization Method	Speed	Oligo to AuNP Density	Au-Nanoprobe Stability	Equipment
Salt-aging	Slow (hours-day)	Low to moderate	Low (at high salt concentration)	Minimal
pH	Moderate (hours)	Moderate	Low (at pH fluctuations)	pH control
Freezing	Moderate	Moderate to high	High	Freezer
Flash synthesis	Very fast (seconds)	Very high	High	High precision required
Microwave heating-dry	Fast (minutes)	Moderate to high	High	Microwave reactor
Evaporative drying	Moderate (minutes hours)	High	High	Simple drying setup

**Table 3 nanomaterials-14-01833-t003:** Examples of the use of Au-nanoprobes for the detection of specific nucleic acid sequences related to diagnostics. This table features experimental work published in the last 8 years, i.e., since 2018.

Spherical AuNPs (Diameter)	Type of Nanoprobe Functionalization and Oligo	Limit of Detection/Specificity	Type of Target	Type of Detection Assay	Authors/Ref	Year	Notes
**20 and 35 nm**	SA (20 nm AuNPs) and/or pH method (35 nm AuNPs), 20 mer HS-oligo	Depends on the type of DNA target (5 ng μL^−1^ for 35 nm AuNPs and 120 mer DNA)	40 and 120-mer Synthetic DNA	Non-cross-linking	Enea et al. [5].	2022	SNP detection
**35 nm**	pH method 16-mer HS-oligo	1.5 ng μL^−1^	Synthetic	Non-cross-linking	Enea et al. [79].	2024	16 bp deletion
**10 and 30 nm**	22 bp HS-oligo	0.313 μM of mutant DNA strand	Synthetic	Non-cross-linking	Ramanathan et al. [80].	2019	EGFR mutation
**13 nm**	pH method thiolated ss DNA	5 nM	Purified samples or whole-cell lysate	Cross-linking approach	Assah et al. [81].	2018	p53 Protein Function assay
**30 nm**	30 min incubation with 4 distinct 20 mer thiol-modified antisense oligonucleotide	0.18 ng μL^−1^	Purified viral RNA	Selective aggregation	Moitra et al. [69].	2020	SARS-CoV-2 Target RNA
**13 nm**	HS-oligos freezing method at −20 °C for 2 h	1 copy/μL limit of detection of pseudoviruses, no cross-reactivity	Plasmid and clinical bio-samples (PCR products)	Cross-linking	Ma et al. [82].	2022	CRISPR-Cas SARS-CoV-2.
**19 nm**	2-thiolated oligos	1 pmol μL^−1^ of 30 bp MERS-CoV	(Synthetic) partial genomic size (30 bp) of upstream E protein gene (upE) and open reading frames (ORFs)	Cross-linking	Kim et al. [42].	2019	Middle East respiratory syndrome coronavirus (MERS-CoV)
**Below 20 nm**	Salt-aging method—3 thiolated SARS-CoV-2 oligonucleotide	0.16 ng μL^−1^	Viral RNA extracted from a nasopharyngeal sample	Combined on visible biosensor color shift (aggregation) and a locally enhanced electromagnetic field and significantly amplified SERS signal	Babadi et al. [83].	2022	Targets four different regions of the viral genome for detection of SARS-CoV-2 and its new variants
**20–25 nm**	Cholesterol modified—thiolated oligonucleotide, long incubation in the presence of NaCl	≥10^3^–10^4^ viral RNA copies/μL	Synthetic samples and samples extracted from infected cells and patients.	Visual colorimetric detection of color change upon aggregation	Diaz et al. [43].	2022	Sequences coding for the RdRp, E, and S proteins of SARS-CoV-2.
**13 nm**	15 mer oligonucleotide	6.83 ng/reaction	Viral RNA—synthetic or extracted from biological samples	Cross-linking	Heidari et al. [27].	2021	Bovine viral diarrhea virus
**13 nm**	15 mer oligonucleotide	44.36 ng/reaction	Viral RNA—synthetic or extracted from biological samples	Non-cross-linking	Heidari et al. [27].	2021	Bovine viral diarrhea virus

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
