# Peer review of "Gold Nanoprobes for Robust Colorimetric Detection of Nucleic Acid Sequences Related to Disease Diagnostics"

_nanomaterials, 2024, doi:10.3390/nano14221833_

Round 1

Reviewer 1 Report

Comments and Suggestions for Authors

This paper offers a clear and concise overview of the use of gold nanoparticles (AuNPs) in colorimetric nucleic acid detection, emphasizing their potential in biosensing applications. The authors address a relevant topic in biosensing, highlighting the advantages of AuNPs, which could appeal to researchers in the fields of biochemistry, nanotechnology, and diagnostics. It effectively outlines the challenges faced in practical applications, which adds depth to the discussion and suggests areas for future research.

Also, the authors could benefit from a bit more detail on how these factors interact or affect assay performance. Similarly, briefly mentioning specific applications or studies that have successfully utilized AuNPs in complex biological matrices could strengthen the review's context and impact. A sentence or two on potential future research directions or innovations would provide a more forward-looking perspective.

The paper seems suitable for publication, especially if it thoroughly analyzes the outlined challenges and potential solutions in subsequent sections. Ensuring that the review is well-structured, includes comprehensive literature citations, and offers insightful discussions on the implications of the findings will enhance its overall quality. In summary, with minor enhancements for specificity and contextual depth, this review could make a valuable contribution to biosensing and gold nanoparticle research.

Author Response

Thank you very much for taking the time to carefully read our review paper. We were extremely pleased for your positive opinion and for the valuablemproving suggestions. Please find our response below and the corresponding revisions highlighted in yellow in the re-submitted file.

Comment 1: Also, the authors could benefit from a bit more detail on how these factors interact or affect assay performance.

Response 1: We thank the reviewer for taking their time to carefully read our review paper and we very much agree with this comment. To explain in more detail how different factors interact or affect assay performenace, we have added to the “5. Challenges and Opportunities” section a new subsection on aggregation x non-aggregation-based methods of detection where many parameters need to be taken in account. Answering this comment, we have added the following sentence, right after that explanation:

By addressing these key elements, researchers can achieve robust, reproducible, and diagnostically relevant results tailored to specific applications, whether in infectious disease detection or broader diagnostic use.

Comment 2: Similarly, briefly mentioning specific applications or studies that have successfully utilized AuNPs in complex biological matrices could strengthen the review's context and impact.

Response 2: We thank the reviewer for this very relevant comment. As examples of successful use of AuNPs in complex biological matrices are lacking, we added mitigation measures to the saliva example presented on the “5. Challenges and Opportunities” section. In the same paragraph, we also added a type of application in which direct analysis would be a great facilitator of the assay, namely miRNA detection in blood:

Authors used protein extraction processes to mitigate these effects [96]. In another example in which the direct use of a biological sample would be helpful by avoiding a prior extraction step, miRNAs as biomarkers in osteoarthritis could only be effectively detected using a cross-linking method, in the patient plasma extracted form, while plasma contents induced undesired AuNP aggregation [97].

We have also used this insightful comment to add details to sample composition mentioned in Table 3, for references [69], [83], and [27], underscoring their origin as “extracts from biological samples”.

Comment 3: A sentence or two on potential future research directions or innovations would provide a more forward-looking perspective.

Response 3: We thank the reviewer for taking their time to read with detail our review paper and we very much agree with this comment. Therefore, we have added the following sentences to the “5. Challenges and Opportunities” section:

Future research on gold nanoparticle-based nucleic acid detection assays could explore advanced functionalization strategies to improve the stability and selectivity of these nanoprobes in complex biological samples. Incorporating machine learning algorithms to fine-tune nanoparticle properties may also enable color multiplexing and enhance the simultaneous detection of multiple targets, expanding the capabilities of point-of-care diagnostics.

Reviewer 2 Report

Comments and Suggestions for Authors

The review manuscript "Gold nanoprobes for robust colorimetric detection of nucleic acid sequences related to disease diagnostics" describes a selection of important approaches applicable in a detection of nucleic acids in samples using plasmic properties of metallic nanoparticles, mainly gold. The text is well structured and inserted figures nicely designed, all in one theme. The text is understandable and statements are supported by relevant references. My only comments are towards the interconnections between chapters and authors critical point of view over them. I suggest the authors to include also potowvoves and negatives of the described approaches and highlight their ideal use and inform the reader about possible problems with their implementation. Especially for the chapters describing aggregation x non-aggregation based methods of detection where many parameters need to be taken in account to make these approaches reliable. 

Author Response

Thank you very much for taking the time to carefully read our review paper. We were extremely pleased for your positive opinion and for the valuable improving suggestions. Please find our response below and the corresponding revisions highlighted in green in the re-submitted files.

Comment 1: I suggest the authors to include also pitfalls and negatives of the described approaches and highlight their ideal use and inform the reader about possible problems with their implementation. Especially for the chapters describing aggregation x non-aggregation-based methods of detection where many parameters need to be taken in account to make these approaches reliable.

Response 1: We thank the reviewer for taking their time to read with detail our review paper and we very much agree with this comment. Therefore, we have added the following analysis to the “5. Challenges and Opportunities” section:

“Crosslinking and non-crosslinking approaches each offer unique advantages and challenges for AuNP-based colorimetric detection. Crosslinking assays, involving two complementary Au-nanoprobes that aggregate in the presence of a target, provide high specificity and can reach attomolar sensitivity with amplification [31, 32]. However, they are slower, require strict temperature control, and show reduced sensitivity without amplification. Non-crosslinking assays, using a single Au-nanoprobe, are quicker and simpler, making them ideal for high-concentration targets and point-of-care use, but are sensitive to ionic strength adjustments, leading to variability and lower sensitivity at low concentrations. Careful optimization of parameters like nanoprobe functionalization, salt concentration, and temperature control is essential to ensure reliable performance across both methods [27, 29]. Furthermore, detection and interpretation techniques like UV/Vis spectroscopy and nanoparticle tracking analysis (NTA) help to quantify aggregation accurately, boosting result accuracy.”

Reviewer 3 Report

Comments and Suggestions for Authors

This review focuses on gold nanoparticles functionalized with oligonucleotides that are used to detect nucleic acid sequences related to diseases by means of colorimetry. The first part details the principle of the detection which is based on the change of color of the nanoparticles upon aggregation and the different methods that are used to induce the aggregation. The second part describes the main functionalization methods and the parameters that can be tuned to optimize the detection. The third part gives some specific examples of the use of these Au-Nanoprobes for disease diagnostics and the last part summarizes the challenges and opportunities in the field.

In believe this review is of interest but needs some improvement before publication

1) a short paragraph on the gold NP synthesis with the most relevant references is missing.

2) Table 1 for the dehydration method, the authors mention under equipment "required precision". It is not clear what they mean here.

3) page 11, first paragraph: the author mentions the "flash synthesis" technique that was not described in the previous paragraphs.

4) page 14, 2nd paragraph: the authors should explain more clearly why a closer proximity of the DNA probe to the gold surface improves hybridization.

5) page 15, 2nd paragraph: the authors should also explain why secondary structures should be avoided to obtain a reliable and sensitive detection.

6) page 16, 2nd paragraph: the autors explain that in the non-cross linking technique, "an oligonucleotide density gretaer than 34 pmol/cm2 imparts long-term stability with red color even at high concentration. I believe here the reliability of the sensor is questioned (with a false negative behavior) rather than its stability.

Comments on the Quality of English Language

The quality of English language can be improved. Some words are missing and the meaning of some sentences is not clear. Some typos should also be corrected.

1) Page 7, 3rd line of paragraph 3: "characteristics of the Au nanoprobes the specific sequences...." (the meaning of the sentence is not clear)

2) page 8, lines 4-5: non-cross linking approach is repeated twice.

3) page 9, line 8: "consistent functionalization of (??) with..."

4) page 11, 1st paragraph, line 12: "the method was used successful used..."

5) page 11, 2nd paragraph, line 5: "It can be successful used"

6) page 11, 2nd paragraph, line 9; "oligonucelotides"

In addition, the numbering of the paragraphs in part 2 is wrong (1.1 shoiuld be 2.1 and so on)

Author Response

Thank you very much for taking the time to review this manuscript, and for valuable improving suggestions. Please find the detailed responses below and the corresponding revisions/corrections highlighted in blue in the re-submitted files

Comment 1: a short paragraph on the gold NP synthesis with the most relevant references is missing.

Response 1: We thank the reviewer for pointing this out. Therefore, we have introduced the following two text blocks, pertaining to the synthesis of spherical and non-spherical AuNPs in section “2. AuNPs optical properties and their use in colorimetric detection”.

Spherical AuNPs display a great variety of synthesis methods resulting in colloids with uniform shapes and low dispersion in size, providing reproducible optical properties  [14,15]. The most widely used spherical AuNPs synthesis methods, especially for smaller AuNPs as for the 15 nm ones, are based on trisodium citrate as both reducing agent of a gold salt and a capping agent for the growing AuNPs [14]. For larger diameters AuNPs (e.g., 40 nm), the seed-mediated growth method allows precise control over particle size by adjusting reaction conditions [15]. Larger AuNPs also have higher extinction coefficients, being more intensely colored (Fig. 2A). Non-spherical AuNPs, such as nanorods and nanostars, display multiple LSPR bands due to their distinct shapes, leading to more complex spectra with overlapping peaks. For non-spherical AuNPs, successful synthesis methods were described in the literature for gold nanostars [16], gold nanotriangles [17] and gold nanorods [18] or other shapes [19,20].

Comment 2: Table 1 for the dehydration method, the authors mention under equipment "required precision". It is not clear what they mean here.

 Response 2: Thank you for pointing this out. To clarify this question, we have replaced “required precision” by high precision required”.

Comment 3: page 11, first paragraph: the author mentions the "flash synthesis" technique that was not described in the previous paragraphs.

Response 3: Thank you for detecting this inconsistency. In fact, the “Instant dehydration method” described in section 3.2.4. and the “Dehydration” method mentioned in Table 1, correspond to the “Flash synthesis” method, so we changed the name accordingly in both instances.

Comment 4: page 14, 2nd paragraph: the authors should explain more clearly why a closer proximity of the DNA probe to the gold surface improves hybridization.

Response 4: Thank you for pointing this out, in fact that paragraph needs rewriting for clarity. The determinant factor for improved hybridization is the shorter spacer and not the closer proximity to the gold surface:

“The preference for the shorter length spacer is associated to a better hybridization efficiency, enhancing the sensitivity of detection assays, and resulting in stronger and more immediate signal changes upon hybridization.”

Comment 5: page 15, 2nd paragraph: the authors should also explain why secondary structures should be avoided to obtain a reliable and sensitive detection. 

Response 5: Thank you for raising this matter, needing clarification. We have added the following text to sub-section “3.4. Composition and length of the recognition sequence”:

“The presence of secondary structure introduces an additional thermodynamic barrier to hybridization, which is especially significant at low ionic strengths [68]. In fact, secondary structures can prevent proper proximity-induced changes, which are crucial for clear signal development.”

Comment 6: page 16, 2nd paragraph: the authors explain that in the non-cross linking technique, "an oligonucleotide density greater than 34 pmol/cm2 imparts long-term stability with red color even at high concentration. I believe here the reliability of the sensor is questioned (with a false negative behavior) rather than its stability.

 Response 6: Thank you for pointing this out, the paragraph was rewritten for clarity:

“34 pmol cm2 imparts long-term stability to the Au-nanoprobe with red color even at high salt concentration,(…)”

Response to Comments on the Quality of English Language

Point 1: The quality of English language can be improved. Some words are missing, and the meaning of some sentences is not clear. Some typos should also be corrected.

1) Page 7, 3rd line of paragraph 3: "characteristics of the Au nanoprobes the specific sequences...." (the meaning of the sentence is not clear). A comma was missing. Corrected

2) page 8, lines 4-5: non-cross linking approach is repeated twice. Repetition eliminated.

3) page 9, line 8: "consistent functionalization of (??) with..." ”with” was eliminated. Corrected.

4) page 11, 1st paragraph, line 12: "the method was used successful used...". Replaced by “The method was successfully used”. Corrected

5) page 11, 2nd paragraph, line 5: "It can be successful used". “successful” replaced by “successfully”. Corrected

6) page 11, 2nd paragraph, line 9; "oligonucelotides". Replaced by “oligonucleotides”. Corrected.

7) In addition, the numbering of the paragraphs in part 2 is wrong (1.1 should be 2.1 and so on). All corrected.

Response 1: We thank the reviewer for noting this important aspect and have made a full analysis of our manuscript for English language quality. Sentences that were re-written for reading clarity and noted typos, are in red lettering in the manuscript.

Round 2

Reviewer 3 Report

Comments and Suggestions for Authors

I thank the authors to have addressed all my suggestions and corrections. I now believe the article is suitable for publication in Nanomaterials.